# Maternal Resources, Pregnancy Concerns, and Biological Factors Associated to Birth Weight and Psychological Health

**DOI:** 10.3390/jcm10040695

**Published:** 2021-02-10

**Authors:** David Ramiro-Cortijo, María de la Calle, Andrea Gila-Díaz, Bernardo Moreno-Jiménez, Maria A. Martin-Cabrejas, Silvia M. Arribas, Eva Garrosa

**Affiliations:** 1Department of Physiology, Faculty of Medicine, Universidad Autónoma de Madrid, C/Arzobispo Morcillo 2, 28029 Madrid, Spain; dramiro@bidmc.harvard.edu (D.R.-C.); andrea.gila@uam.es (A.G.-D.); 2Department of Medicine, Beth Israel Deaconess Medical Center, Harvard Medical Center, 330 Brookline Avenue, Boston, MA 02215, USA; 3Obstetrics and Gynecology Service, Hospital Universitario La Paz, Paseo de la Castellana 261, 28046 Madrid, Spain; maria.delacalle@uam.es; 4Pharmacology and Physiology PhD Degree, Escuela de Doctorado, Universidad Autónoma de Madrid, Ciudad Universitaria de Cantoblanco, 28049 Madrid, Spain; 5Department of Biological & Health Psychology, Faculty of Psychology, Universidad Autónoma de Madrid, C/Ivan Pavlov 6, 28049 Madrid, Spain; bernardo.moreno@uam.es; 6Department of Agricultural and Food Chemistry, Faculty of Science, Universidad Autónoma de Madrid, C/Francisco Tomás y Valiente 7, 28049 Madrid, Spain; maria.martin@uam.es

**Keywords:** maternal depression, family–work conflict, leukocytes, optimism, polyphenols, resilience

## Abstract

Cognitive maternal adaptation during pregnancy may influence biological variables, maternal psychological, and neonatal health. We hypothesized that pregnant women with numerous general resources and less negative emotions would have a better coping with a positive influence on neonatal birth weight and maternal psychological health. The study included 131 healthy pregnant women. A blood sample was obtained in the first trimester to assess biological variables (polyphenols, hematological and biochemical parameters). Psychological variables (negative affect, anxiety, optimism, resilience, family–work conflicts, pregnancy concerns, general resources, and life satisfaction) were evaluated at several time points along gestation, and birth weight was recorded. Hierarchical linear regression models were used to associate the above parameters with maternal psychological outcome at the end of gestation (depression, resilience, and optimism) and neonatal outcome (birth weight). Maternal depression was associated with leukocytes (β = 0.08 ± 0.03, *p*-value = 0.003), cholesterol (β = 0.01 ± 0.002, *p*-value = 0.026), and pregnancy concerns (β = 0.31 ± 0.09, *p*-value = 0.001). Maternal resilience was associated with leukocytes (β = −0.14 ± 0.09, *p*-value = 0.010) and life satisfaction (β = 0.82 ± 0.08, *p*-value = 0.001), and maternal optimism was associated with polyphenol levels (β = 0.002 ± 0.001, *p*-value = 0.018) and life satisfaction (β = 0.49 ± 0.04, *p*-value = 0.001). Birth weight was associated with maternal resilience (β = 370.2 ± 97.0, *p*-value = 0.001), red blood cells (β = 480.3 ± 144.4, *p*-value = 0.001), and life satisfaction (β = 423.3 ± 32.6, *p*-value = 0.001). We found associations between maternal psychological, blood variables, and birth weight and maternal depression. This study reveals the relevance of psychological health during pregnancy for maternal and neonatal outcome, and it emphasizes the need to consider it in preventive policies in the obstetric field.

## 1. Introduction

Pregnancy is a major life transition demanding adjustment of many situations [1]. A woman’s skills to adjust to the stressors and challenges of pregnancy have effects on both her physical and psychological health. In turn, this plays an important role in the health of the developing infant [2]. A more complex understanding of psychological adaptation during pregnancy and its effects on health outcomes requires consideration as one of the many factors that may influence prenatal adaptation.

Research on pregnancy health is of utmost importance from the clinical and social point of view. A healthy pregnancy not only ensures maternal well-being but also has an impact on health of the future generations. Low birth weight (LBW; birth weight < 2500 g) is a key parameter, since it is a representation of the quality of intrauterine life, and it is a critical factor for the future health of the individual [3]. It is estimated that each year, 16% of births are LBW, and the rate is increasing, mainly due to premature births. In low-income countries, LBW is related to poverty and early pregnancy. In high-income societies, it is mainly linked to the increase in childbearing age, which is in turn associated with obstetric complications and infertility and need of assisted reproduction techniques (ART), leading to multiple pregnancies and LBW [4]. Despite considerable research attention, the knowledge on the etiology of LBW is still incomplete [5]. Research has been largely focused on physical health. However, it is also important to consider psychological well-being, such as the absence of depression of the mother and her resilience, which are related to a healthy motherhood and the well-being of the infant [6].

Pregnancy could be a burden due to lack of adequate psychosocial resources and the presence of additional stressors, such as family–work conflict. The conservation of resources model (COR) [7,8] has been previously applied to examine the stressful influences of women’s multiple roles during pregnancy [9]. For many women, pregnancy is neither an idyllic gestational period nor no-stressor situation [10]. According to COR, stress causes not only a failure of maternal psychological resources—such as resilience, optimism, and social support—to maintain control of the situation but also represents a threat to lose life satisfaction. The loss of maternal resources may exert a negative influence on pregnancy health. On the other hand, it has been demonstrated that the response to a stressful situation can be modulated by factors such as resilience and coping, which can improve adaptive physiological and emotional responses to stressors, with a positive influence on health [11,12].

Psychological factors are closely linked to biological variables. Firstly, psychologically disruptive situations may alter healthy habits. In particular, dietary pattern may be affected with important health consequences; for example, inadequate diet may lead to an insufficient intake of fruits and vegetables, and thus exogenous antioxidants. We have demonstrated the relationship between low maternal antioxidant status and development of pregnancy complications [13]. We also evidenced a direct relationship between fruit and vegetable intake and maternal plasma polyphenol content, which are the major antioxidants in foods of vegetable origin [13]. The importance of diet in pregnancy outcome is also supported by the association between pre-pregnancy consumption of a Mediterranean-style dietary (rich in fruits and vegetables) and lower risk of adverse obstetric outcome [14]. Furthermore, there is also evidence that psychological variables may affect hematological parameters, and it has been described that stressful situations are related with increased leukocytes [15]. An altered balance of immune cells in pregnancy may also be related to inflammatory-related alterations and affect maternal health [16].

We hypothesized that pregnant women with abundant general resources and less factors related to negative emotions would have a better psychological adaptation to pregnancy with beneficial consequences for maternal and neonatal outcome. To test this hypothesis, we have analyzed in a cohort of pregnant women the evolution of variables related to negative emotions (negative affect and anxiety), psychological stress (family–work conflict and pregnancy concerns), and psychosocial resources (general resources and life satisfaction). Secondly, we have assessed the impact of these variables on neonatal birth weight and on maternal depression, resilience, and optimism at the end of gestation. Thirdly, we have analyzed some biological parameters: plasma polyphenol levels as major antioxidants reflecting an adequate consumption of fruits and vegetables [13], and hematological parameters, which have been previously demonstrated to be influenced by psychological stress [15] and are altered in adverse obstetric outcomes [17].

## 2. Materials and Methods

### 2.1. Study Design and Participants

This non-interventional and observational study was approved by the Universidad Autónoma de Madrid (Spain) and Hospital Universitario La Paz (HULP, Madrid, Spain) Research Ethics Committee (Refs. 2013-10 and PI-1490, respectively).

A total of 131 healthy pregnant women were enrolled at the Obstetrics and Gynecology Service from HULP from January 2014 to June 2015. Women entered the study confidentially and voluntarily signed an informed consent to participate in the study. Exclusion criteria were women with previous immunological/cardiovascular disease (any type of anemia, diabetes mellitus, body mass index ≥ 30 before pregnancy, and hypertension) or risk factors (smoking habits or regular medications). The inclusion criteria were good comprehension of the Spanish language.

Women entered the study at nine weeks of gestation and were followed up at HULP until labor. At the beginning of the study, each participant completed a sociodemographic survey (educational level: none/pre-university/university, civil status: single/married and work situation: with/without employment). At ten weeks of gestation, two blood samples of 4 mL each were extracted, in fasting state, from 8:00 to 9:00 a.m. by venipuncture in heparin tubes (BD Vacutainer®, Oakville, ON, Canada), following the protocols established by the medical staff. In addition, each woman self-reported at each trimester of gestation (nine, twenty-four, and thirty-six weeks) the psychological questionnaire (see below).

The following clinical variables were collected from the medical records: maternal age, parity, type of pregnancy (single/twin; determined by echography at ten week of gestation); type of gestation (spontaneous or ART); gestational age (weeks of gestation); miscarriage (yes/no); route of delivery (vaginal/C-section); and neonate birth weight in grams (precision scale range of 0 to 20 Kg; Kern & Sohn GmbH, Balingen, Germany).

### 2.2. Blood Variables

#### 2.2.1. Hematological and Biochemical Parameters

A blood sample was used to assess red blood cells (10^6^/mL), platelets (10^3^/mL), leukocytes (10^6^/mL), glucose (mg/dL), total cholesterol (mg/dL), and triglycerides (mg/dL) by the Clinical Laboratory Service of HULP.

#### 2.2.2. Polyphenol Levels

The second blood sample was used to obtain plasma by centrifugation (2100× *g*, 15 min at 4 °C) within a maximum of 2 h after extraction. Thereafter, Folin–Ciocalteu assay, which was modified to remove protein interference, was used to assess food-derived antioxidants, mainly phenolic compounds and ascorbic acid [18]. Absorbance was measured at 760 nm, and the results were expressed as mg gallic acid equivalent per liter (mg GAE/L).

### 2.3. Psychological Variables

#### 2.3.1. Negative Emotions

Negative emotions were assessed using standardized scales of the Positive and Negative Affect Schedule (PANAS) [19] and the Hospital Anxiety and Depression scale (HAD) [20].

The PANAS is one of the most widely used scales to measure emotion. We have used 10 items for measuring negative affect (e.g., upset, afraid) in the general context. Each item is rated on a five-point Likert scale, ranging from 1 (very slightly or not at all) to 5 (extremely), to measure the extent to which the effect has been experienced in a specified time frame. We used the Spanish version of PANAS [21] and collected responses in the first trimester of gestation. Our negative affect scale had an internal consistency of 84%.

The HAD evaluates the cognitive, emotional, and behavioral responses of anxiety and depression in general population. This scale consists of 14 items with 4 Likert response options and provides separate scores using two subscales: anxiety and depression. We used the Spanish version of HAD [22]. Our anxiety and depression scales had an internal consistency of 73% and 80%, respectively. Anxiety was measured in the first trimester, and depression was measured at the end of the pregnancy.

#### 2.3.2. Personal Resources

These variables of personality were evaluated using standardized scales of dispositional optimism by Life Orientation Test Optimism (LOT) [23]. This scale consists of five items. Subjects rate each item on a 5-point scale, ranging from 1 (strongly disagree) to 5 (strongly agree). We used the Spanish version of LOT [24]. Our dispositional optimism scale had an internal consistency of 64% and was completed at the first and the end of pregnancy.

Another personal resource test used was the Resilience scale [25], which was completed at the start and the end of pregnancy. The Resilience scale is composed of 6 items of closed type, which are constructed according to a Likert scale of 7 alternatives, ranging from 1 (totally disagree) to 7 (totally agree). We used the individual Spanish version of this Resilience scale, which was adapted to the Spanish-speaking population [26]. In our study, this scale had an internal consistency of 66%.

#### 2.3.3. Psychosocial Stress

This aspect was evaluated by two ad hoc scales: family–work conflict and pregnancy concerns. Both were measured at the second trimester of pregnancy. The family–work conflict test has the purpose of evaluating discrepancies between the personal/family environment and the work context. A poor fit between these two areas of life can indicate a generalized psychological stress. The scale consists of 6 items with a Likert response from zero (never) to three (always). Sample items included, “*The demands of my work interfered with my home and family life*” and “*The amount of time my job takes up makes it difficult to fulfill family responsibilities*”. This scale had an internal consistency of 88%.

The pregnancy concerns scale was elaborated according to the current concerns model of E. Klinger [27], which is aimed at assessing the particular concerns derived from the pregnancy process. The majority of questionnaires assessing psychosocial stress have been built for the general population [28,29,30], although there are some adaptations of these scales to pregnancy [31]. We used a variation, consisting of a short questionnaire focused on the child and mother health, adapted to the current study cohort. This test had 10 items and a response scale from zero (none) to three (very). These scales were specifically designed for this study and include concerns about the time of labor, body image after childbirth, and problems with the partner or economic difficulties. Items included in the scale to determine pregnancy worries were “*The newborn health*” or “*The medical attention during labor*”. Our pregnancy concerns scale had an internal consistency of 70%.

#### 2.3.4. General Resources

The general resources questionnaire derives from Antonovski’s health psychology model [32], which introduces the personal general resources as an element of resistance to disease. This model has been proposed in other contexts [33,34,35]. In the absence of specific questionnaires, and the importance of the construct in the mothers’ health model, we opted for its elaboration and psychometric control. We analyzed general resources in our pregnant cohort using standardized scales of general resources (i.e., economic, social, and family resources), consisting of 9 items with a response scale from 1 (strongly agree) to 5 (strongly disagree). Example of items were “*I do not have important economic problems*” and “*I have a great social network*”. This scale had an internal consistency of 86% and was measured at the second trimester of pregnancy.

The life satisfaction scale [36] had 5 items constructed according to a Likert scale of 7 alternatives ranging between 1 (totally disagree) and 7 (totally agree). We used the Spanish version of life satisfaction for pregnant women [37]. Our life satisfaction scale had an internal consistency of 85% and was completed during the second trimester of pregnancy.

### 2.4. Statistical Analysis

Quantitative variables are shown as mean ± standard deviation (SD) and correlations were tested by Pearson’s coefficients. Qualitative variables are described as relative frequency, and chi-square was used to assess their association.

Pregnant women completed the psychological questionnaires at the three time points (first trimester: negative affectivity, anxiety, optimism, and resilience; second trimester: family–work conflict, pregnancy concerns, general resources and life satisfaction; third trimester: depression, resilience and optimism). The temporal aspect of this research helped guarantee that the effects of these variables were not attributable to methodological artefacts inherent to cross-sectional designs, which is a common methodological bias [38]. To test the hypotheses of the relationship between proposed variables and primary outcomes at the end of pregnancy (birth weight, depression, resilience, and optimism), hierarchical linear regression analyses were conducted. Maternal age, educational level, civil status, work situation, ART, parity, and previous miscarriages were entered in the analyses as maternal variables. In the second step, we included variables measured at first trimester of gestation: negative emotions (negative affect and anxiety), personal resources (optimism and resilience), and blood parameters. In the last step, the variables measured at second trimester of gestation were introduced: family–work conflict, pregnancy concerns, general resources and life satisfaction. Data were checked for collinearity using tolerance and the variance inflation factor. Results were expressed as estimated beta (β) coefficients, standard error (SE), and associated *p*-value. In addition, adjusted explained variance (R^2^) in each step was reported. Statistical analysis was performed with SPSS Statistic version 25.0 (IBM Company, Armonk, NY, USA).

## 3. Results

Participation rates were 91.5% (119/131) at first trimester, 79.5% (104/131) at second trimester, and 65.5% (86/131) at the end of gestation. Follow-up loss of participants was due to abortions, preterm labor, or that the woman forgot to complete the questionnaires. In the adjusted lineal regression models, only women with all observations were included.

The mean maternal age was 34.4 ± 4.5 years old, 69.7% (91/131) of the women had university studies, 71.6% (94/131) were married, and 90.4% (118/131) were employed. The mean parity was 1.5 ± 0.5, and the mean gestational age was 37.4 ± 2.5 weeks of gestation. The rate of twin pregnancies was 51.8% (68/131) and the ART-derived gestation rate was 36.9% (48/131). Both variables were significantly associated (*p*-value = 0.001). C-section was 48.5% (64/131) and male newborn rate was 56.9% (75/131). Average birth weight was 2717.0 ± 672.0 g.

Maternal blood parameters measured at the first trimester were all within normal ranges: red blood cells = 4.3 ± 0.4 × 10^6^/mL, platelets = 251.7 ± 55.9 × 10^3^/mL, leukocytes = 8.1 ± 1.9 × 10^6^/mL, glucose = 93.9 ± 31.4 mg/dL, total cholesterol = 174.5 ± 27.8 mg/dL, triglycerides = 92.0 ± 35.7 mg/dL. The average maternal plasma polyphenols were 291.5 ± 59.7 mg GAE/L. Descriptive data and correlation between psychological and biological variables, and primary outcomes (birth weight, depression, resilience, and optimism) are shown in Table 1.

The scores and correlations between psychological variables used in the present study are shown in Appendix A.

Table 2 shows the associations between the analyzed variables and birth weight as a parameter indicative of neonatal outcome. ART (as type of gestation) exhibited a significant and negative association. We aimed to test the role of maternal social variables on maternal and neonatal outcomes. Due to the high collinearity between ART and twin pregnancies, ART was included in the models.

We found positive and significant associations between birth weight and resilience score (β = 370.2 ± 97.0; *p*-value = 0.001) and red blood cells (β = 480.3 ± 144.4; *p*-value = 0.001) in the first trimester of pregnancy, and with life satisfaction in the second trimester (β = 423.3 ± 32.6; *p*-value = 0.001; Table 2).

Table 3 shows the associations between social, psychological, and biological variables and maternal psychological outcome at the end of pregnancy.

Maternal depression at the end of pregnancy was negative and significantly associated with maternal age (β = −0.04 ± 0.02, *p*-value = 0.019), while positive associations were found with educational level (β = 0.33 ± 0.14, *p*-value = 0.023), leukocyte levels (β = 0.08 ± 0.03, *p*-value = 0.003), and total cholesterol (β = 0.01 ± 0.002, *p*-value = 0.026) in the first trimester. Pregnancy concerns in the second trimester of pregnancy were also significantly and positively associated with depression (β = 0.31 ± 0.07, *p*-value = 0.001; Table 3).

Maternal resilience at the end of pregnancy showed significant and positive associations with maternal age (β = 0.16 ± 0.04, *p*-value = 0.001), married status (β = 0.89 ± 0.31, *p*-value = 0.006), and resilience (β = 0.76 ± 0.16, *p*-value = 0.001), while a negative and significant association was found with leukocytes at the beginning of pregnancy (β = −0.14 ± 0.09, *p*-value = 0.010). Life satisfaction in the second semester of pregnancy was also significant and positively associated with resilience (β = 0.82 ± 0.08, *p*-value = 0.001; Table 3).

Maternal optimism at the end of pregnancy showed significant and positive associations with maternal age (β = 0.10 ± 0.02, *p*-value = 0.001), married status (β = 0.41 ± 0.17, *p*-value = 0.018), optimism (β = 0.57 ± 0.10, *p*-value = 0.001), resilience (β = 0.10 ± 0.08, *p*-value = 0.024), and plasma maternal polyphenol levels at the beginning of pregnancy (β = 0.002 ± 0.001, *p*-value = 0.018). Life satisfaction in the second trimester of pregnancy was also significant and positively associated with optimism (β = 0.49 ± 0.04, *p*-value = 0.001; Table 3).

A summary of the main associations found in the present study is shown in Table 4.

## 4. Discussion

In this study, we aimed to assess the possible relationship between maternal psychological and biological factors and maternal and neonatal pregnancy outcomes. We have implemented an integrative model that includes variables related to obstetric history, biological factors, and psychosocial variables in the context of pregnancy. We found several relationships that evidence the importance of psychosocial factors, for pregnancy outcome and their relationship with biological variables. With respect to maternal psychological outcome, we evidenced an association between leukocytes and cholesterol levels at the beginning of the pregnancy and depression and resilience at the end of pregnancy. In addition, pregnancy concerns in the second trimester were also associated with depression. Furthermore, resilience and optimism in the first trimester and life satisfaction in mid-pregnancy were related to resilience and optimism at the end of pregnancy. Similarly, maternal plasma polyphenols at the beginning of gestation were associated with optimism at the end of pregnancy. In our models, the sociodemographic predictive factors were maternal age, educational level, and civil status. Regarding neonatal outcome, we found that the use of ART and red blood cells, resilience, and life satisfaction were linked with birth weight.

We explored two primary outcomes, neonatal and maternal psychological health. Regarding neonatal health, we assessed the impact of the variables on birth weight, as a representative of the quality of intrauterine life. As expected, ART was a negative influence on birth weight. This is not inherent to the reproduction technique itself, but to the fact that women on ART usually have multiple pregnancies and lower birth weight [39]. Regarding the impact of maternal psychological well-being on infant weight, we used the COR model [7,8] to understand the role of resource loss and gain during pregnancy. In the present study, it was found that family–work conflict was correlated with LBW, suggesting the important influence of stressful working condition on pregnancy outcome. Birth weight is a key clinical parameter and an important predictor of adulthood health [4]. LBW increases the risk of infant mortality, impairs neurodevelopment in infants who survive [40], and predicts vulnerability to cardiovascular and metabolic disorders in adulthood [41]. These findings are consistent with the evidence that experiences of family–work conflict transmit stress in both the family and work areas. We suggest that stress transmission has a negative impact on maternal health and neonatal outcome. Family–work conflict, as a stressor, probably leads to exhaustion of cognitive and emotional resources. Such resources are essential in controlling and handling demanding situations [42]. Therefore, this psychological stress and resources loss could have a negative impact not only maternal health but also on infant weight. An association between birth weight and red blood cells was also found. Low levels of erythrocytes may reflect inadequate diet [43], and anemia during pregnancy has been associated with LBW infants [44]. Since birth weight was negatively associated with family–work conflicts, we suggest that stress factors during pregnancy may negatively influence self-care of the woman.

Additionally, we examined specific variables, which are related to maternal psychological health and well-being. Research on stress and pregnancy has shown that domain-specific stresses can differentially predict pregnancy outcomes [2]. Regarding social aspects, we found that educational level was associated with depression. A possible explanation is that women with higher educational level may have more information about possible pregnancy problems. In fact, a relationship between depression and pregnancy concerns was also observed in the present study. However, the age of maternity was negatively associated with depression. It is possible that mature woman can cope better with their decision to postpone pregnancy. Conversely, some studies have evidenced that advanced mother age may be a factor in the development of depression after childbirth [45]. This suggests that the influence of maternal age on post-partum depression is complex and may be modulated by social factors. Among other factors that may explain our results, we suggest that older women have a more stable family core. In fact, married status was positively associated with resilience and optimism. In our cohort, a more stable economic position may also contribute to this association.

We evidenced that depression was also correlated with previous anxiety in the mother. Women with depression often experience symptoms such as nervousness, irritability, and sleep and concentration problems. Individuals who develop depression frequently have a previous history of anxiety and although there is no evidence one disorder causes the other, both are often associated [46].

We also explored some biological factors and their possible relationship with psychological variables. In our study, we found that depression at the end of pregnancy was associated by high leukocytes levels in the first trimester and by pregnancy concerns in the second trimester of gestation. Some studies have shown the presence of an inflammatory response in the depression process [15,47]. Alterations in leucocytes may implicate an early inflammatory response [48]. Inadequate inflammatory responses are also critical for pregnancy outcome. Whether an early elevation of leucocytes is associated with a pro-inflammatory environment and if this can play a role in depression development warrants further analysis. It is also important to highlight that inflammation includes a variety response of multiple type of cells and molecules, and this aspect of research deserves additional studies. It is important to note that leukocytes are routinely measured in pregnant women, and if a relationship is found, it could be a potential early biomarker of depression development in pregnancy. Our findings suggest that increased leukocyte levels at the beginning of a healthy pregnancy could be involved in the mother perception of maternal depression and resilience. A meta-analysis suggested that depression was associated with immune activation of acute phase response, which was led by impairments in Natural Killer (NK)- and T-cell-mediated functions [49]. However, the cause–effect is not entirely clear. Our model suggests that an early leukocyte elevation could potentially predict subsequent mood alterations. Unfortunately, the different leukocyte populations were not collected in our study, and this hypothesis should be further explored. These results evidence the synergic interaction between biological and psychological factors in the pregnancy context.

We also found a positive association between cholesterol levels and depression. This relationship has been poorly described. Despite the fact that cholesterol levels were within the normal ranges, it is possible that higher levels reflect a less healthy diet, with higher consumption of fats and sugars. In agreement with this hypothesis, Wei and co-workers have demonstrated that maternal stress during early pregnancy is associated with a high-fat and sugar dietary pattern, which may mediate a high birth weight [50]. In this article, the authors did not explore biological pathways, and we propose that maternal plasma cholesterol and triglycerides could be potential biomarkers. A high-fat diet is not only deleterious for the pregnant women but also for the offspring, leading to alterations in the development of the serotonergic system in the fetus, as well anxiety-like behavior in primates [51,52].

Additionally, women who showed low level of concerns about pregnancy also had a trend to have a greater resilience. In agreement, some authors indicate that pregnancy concerns can be considered as an indicator of stress, influencing pregnant women’s health, and it is inversely related to resilience. Resilience is defined as a dynamic process reflecting the capacity of the individual to adapt in response to threats or traumatic experiences [53]. Resilience could be a predictor of life satisfaction and pregnancy concerns. Our data indicate a significant and positive correlation between resilience and life satisfaction. We should consider this relationship as bi-directional; i.e., it is possible that the higher resilience, the higher life satisfaction, and the lower pregnancy concerns. On the other hand, pregnancy concerns might trigger the alarm stress response and result in a large number of physiological effects [54]. Different studies have identified specific worries associated with pregnancy and how pregnancy might trigger them [55]. Our results are in agreement, showing the key role of the variable pregnancy concerns at twenty-four weeks of gestation to explain the resilience of women at the end of pregnancy.

Finally, we analyzed optimism in the first and the last trimesters of pregnancy. As we expected, optimistic women kept this positive aptitude along gestation and maintain a positive feeling also at labor. The optimism may play a role in contributing to an overall healthy pregnancy and the postpartum experience. This process can have numerous benefits not only for women challenged with the risk of postpartum depression but also for their neonates, families, and relationships [56]. Additionally, we found that anxiety was negatively correlated with optimism as a maternal outcome variable. This negative relationship is well known; our study supports this association and adds new information regarding the temporal effect of optimism along pregnancy. We also found that optimism at the end of pregnancy was positively associated with maternal plasma polyphenol levels. Polyphenols are obtained from fruits and vegetables, and their health benefits are related to their antioxidant effects to protect cell damage [57]. In a previous study, we observed a cluster between high intake of fruits and vegetables and plasma polyphenol levels; it was also observed that increased levels of antioxidants were associated with better obstetric outcome [13]. The important role of polyphenols associated with optimism observed in the present study suggest that optimistic women could have a greater control over their diet and possibly on other healthy habits. Optimists display adaptive coping and better self-regulation, both of which are critical to achieve successful behavior change. A better understanding of the relationships between optimism and diet can help explain how this personal attitude may be used as a stimulus for healthy nutrition and well-being [58].

### Study Highlights and Limitations

One of the strengths of the present work was the longitudinal design, which enables establishing the predictors of psychological maternal health and neonatal outcomes. On the other hand, the homogeneity of the sample, including mainly middle class women, may be a weak point of the study, limiting the generalization of our findings. Therefore, it would be interesting to analyze these results in other populations with different sociodemographic patterns and origins such as low-income parents and different cultural groups. Furthermore, it would be interesting to measure the maternal diet pattern since, as we have commented before, polyphenols or even leukocytes could be altered under dietetic conditions. The healthy habits could be another variable to consider in the models.

## 5. Conclusions

This study fills a gap in knowledge regarding the positive role of optimism and resilience at the beginning of pregnancy to maintain these positive psychological variables by the end of gestation. It also proposes that high leukocyte and cholesterol levels may be positively associated with depression and negatively associated with resilience at the end gestation. In addition, pregnancy concerns and life satisfaction could be predictors of depression and resilience/optimism, respectively. This relationship should be further explored, considering resilience a key determining factor modulating life satisfaction and pregnancy concerns. Therefore, routine blood parameters evaluated in pregnancy could be taken into account to include vulnerable women in specific programs to strengthen the necessary personal resources. Finally, it would be important to consider the relationship found between maternal resilience and life satisfaction with birth weight, which is a key factor of infant health. Maternal psychological health is an important sphere with a direct impact on maternal and neonatal outcome and needs to be considered for preventive policies in the obstetric field.

## Figures and Tables

**Table 1 jcm-10-00695-t001:** Correlations between psychological variables and primary outcomes at the end of the study.

	Birth Weight (g)	Depression	Resilience	Optimism
Mean ± SD	2717.0 ± 672.0	0.8 ± 0.5	5.5 ± 1.5	3.4 ± 0.6
*First trimester variables*				
Negative affect	−0.2 *	0.4 *	−0.2 *	−0.2 *
Anxiety	−0.1	0.3 *	−0.2	−0.3 *
Optimism	−0.1	−0.2	0.2 *	0.7 *
Resilience	0.2 *	−0.2 *	0.2 *	0.3 *
Red blood cells	0.1	0.1	−0.1	−0.1
Platelets	−0.2 *	0.03	−0.1	−0.1
Leukocytes	−0.3	0.3 *	−0.3 *	−0.1
Glucose	−0.1	−0.2	0.1	−0.1
Total cholesterol	−0.2 *	0.1	−0.1	−0.03
Triglycerides	−0.3 *	0.2	−0.1	−0.03
Polyphenols	−0.04	0.02	0.02	0.3 *
*Second trimester variables*				
Family–work conflict	−0.2 *	0.2	−0.1	−0.1
Pregnancy concerns	−0.1	0.3 *	−0.1	−0.1
General resources	−0.1	0.2 *	−0.1	−0.2
Life satisfaction	0.3 *	−0.2	0.2	0.2 *

Data show Pearson’s coefficients. * *p*-value < 0.05.

**Table 2 jcm-10-00695-t002:** Lineal regression analysis for birth weight as a dependent variable.

	Birth Weight	*p*-Value
*Step 1—Maternal variables (R^2^ = 0.95; p-value = 0.001)*		
Maternal age	58.5 ± 31.6	0.07
Educational level: University degree	578.7 ± 321.1	0.08
Civil status: married	−16.9 ± 289.5	0.95
Work situation: no employment	−341.6 ± 437.9	0.44
Assisted reproduction techniques	−745.6 ± 346.5	0.038
Parity	142.7 ± 285.2	0.62
Previous miscarriage	235.0 ± 223.3	0.30
*Step 2—First trimester variables (R^2^ = 0.95; p-value = 0.001)*		
Negative affect	−96.7 ± 135.0	0.48
Anxiety	147.7 ± 191.1	0.44
Optimism	−116.8 ± 105.7	0.27
Resilience	370.2 ± 97.0	0.001
Red blood cells	480.3 ± 144.4	0.001
Platelets	−0.13 ± 1.2	0.91
Leukocytes	−49.6 ± 39.4	0.21
Glucose	−0.65 ± 2.2	0.77
Total cholesterol	−0.17 ± 2.6	0.95
Triglycerides	−4.7 ± 2.5	0.06
Polyphenols	−0.73 ± 1.1	0.50
*Step 3—Second trimester variables (R^2^ = 0.94; p-value = 0.001)*		
Family–work conflict	−33.4 ± 92.8	0.72
Pregnancy concerns	69.1 ± 113.1	0.54
General resources	147.1 ± 86.0	0.09
Life satisfaction	423.3 ± 32.6	0.001

Data show estimated beta coefficients ± SE and *p*-values. Adjusted explained variance (R^2^) and associated *p*-values are shown at each step.

**Table 3 jcm-10-00695-t003:** Lineal regression analysis for depression, resilience, and optimism, as dependent variables, at the end of pregnancy.

	Depression	*p*-Value	Resilience	*p*-Value	Optimism	*p*-Value
*Step 1—Maternal variables (Depression: R^2^ = 0.70; Resilience: R^2^ = 0.97; Optimism: R^2^ = 0.97)*						
Maternal age	−0.04 ± 0.02	0.019	0.16 ± 0.04	0.001	0.10 ± 0.02	0.001
Educational level: University degree	0.33 ± 0.14	0.023	−0.61 ± 0.36	0.10	−0.23 ± 0.20	0.25
Civil status: married	0.05 ± 0.12	0.68	0.89 ± 0.31	0.006	0.41 ± 0.17	0.018
Work situation: no employment	0.24 ± 0.23	0.31	−0.37 ± 0.61	0.55	0.02 ± 0.33	0.96
Assisted reproduction techniques	0.21 ± 0.15	0.17	−0.51 ± 0.40	0.21	−0.34 ± 0.22	0.12
Parity	0.16 ± 0.12	0.20	0.31 ± 0.32	0.33	0.06 ± 0.17	0.73
Previous miscarriage	0.08 ± 0.11	0.48	0.51 ± 0.29	0.08	0.11 ± 0.16	0.48
*Step 2—First trimester variables (Depression: R^2^ = 0.74; Resilience: R^2^ = 0.93; Optimism: R^2^ = 0.98)*						
Negative affect	0.20 ± 0.12	0.08	0.14 ± 0.40	0.73	−0.01 ± 0.13	0.91
Anxiety	0.11 ± 0.16	0.48	0.13 ± 0.54	0.81	0.08 ± 0.17	0.66
Optimism	−0.07 ± 0.09	0.45	0.37 ± 0.32	0.25	0.57 ± 0.10	0.001
Resilience	−0.01 ± 0.07	0.86	0.76 ± 0.16	0.001	0.10 ± 0.08	0.024
Red blood cells	−0.04 ± 0.12	0.73	0.05 ± 0.40	0.91	−0.04 ± 0.13	0.75
Platelets	0.0 ± 0.001	0.99	−0.0 ± 0.003	0.86	−0.0 ± 0.001	0.61
Leukocytes	0.08 ± 0.03	0.003	−0.14 ± 0.09	0.010	−0.00 ± 0.04	0.97
Glucose	−0.00 ± 0.002	0.23	0.01 ± 0.01	0.28	0.00 ± 0.002	0.40
Total cholesterol	0.01 ± 0.002	0.026	−0.00 ± 0.01	0.68	0.00 ± 0.002	0.72
Triglycerides	−0.00 ± 0.002	0.15	0.00 ± 0.01	0.99	0.00 ± 0.002	0.71
Polyphenols	0.00 ± 0.001	0.85	0.00 ± 0.003	0.36	0.00 ± 0.001	0.018
*Step 3—Second trimester variables (Depression: R^2^ = 0.70; Resilience: R^2^ = 0.93; Optimism: R^2^ = 0.96)*						
Family–work conflict	0.07 ± 0.08	0.35	0.15 ± 0.25	0.56	0.04 ± 0.11	0.73
Pregnancy concerns	0.31 ± 0.09	0.001	0.12 ± 0.28	0.66	0.21 ± 0.12	0.09
General resources	0.13 ± 0.07	0.05	0.27 ± 0.21	0.20	0.10 ± 0.09	0.27
Life satisfaction	−0.03 ± 0.03	0.29	0.82 ± 0.08	0.001	0.49 ± 0.04	0.001

Data show estimated beta coefficients ± SE. Adjusted explained variance (R^2^). The associated *p*-value = 0.001 in each step.

**Table 4 jcm-10-00695-t004:** Associations between maternal, first and second trimester variables with primary outcomes at the end of pregnancy.

Variables		Birth Weigh	Depression	Resilience	Optimism
Maternal	Maternal age		Negative	Positive	Positive
	EL: University degree		Positive		
	Civil status: married			Positive	Positive
	ART	Negative			
First trimester	Resilience	Positive		Positive	Positive
	Optimism				Positive
	Red blood cells	Positive			
	Leukocytes		Positive	Negative	
	Total cholesterol		Positive		
	Polyphenols				Positive
Second trimester	Life satisfaction	Positive		Positive	Positive
	Pregnancy concerns		Positive		

Educational Level (EL); Assisted Reproduction Techniques (ART).

## Data Availability

The data presented in this study are available on request from the corresponding author.

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
