# Peer review of "Maternal Resources, Pregnancy Concerns, and Biological Factors Associated to Birth Weight and Psychological Health"

_jcm, 2021, doi:10.3390/jcm10040695_

Round 1

Reviewer 1 Report

Thanks for the opportunity to review this manuscript. The topic is interesting, and the strengths of the research include longitudinal design, taking into account the bio-psycho-social nature of the variables, which allows drawing application conclusions.
At the same time, I have a few doubts. I list them below with a recommendation to introduce appropriate changes or clarifications:

  1. The authors state that they applied “(...) two ad-hoc scales, family-work conflict, and pregnancy concerns” (p. 4, lines 172-174). While they quite accurately describe the structure and what is measured by the second tool created for the purposes of this study, the scale for measuring family-work conflict is described with little precision. Please provide more detailed description
    2. Why were the variables entered into the regression equation by time of measurement? It seems more reasonalble to group the variables according to their nature, e.g. medical, related to reproductive experiences (ART, parity miscarriages ...).
    3. The authors state that: “The rate of twin pregnancies was 51.8% (...)" (p.5, lines 231-232) -  one of the main variables representing the neonatal outcome is the child's birth weight. At the same time, more than half of the pregnancies are twin pregnancies in which the birth weight is usually lower. In this situation, it seems appropriate to perform a regression analysis separately for single and twin pregnancies.
    4. How did the Authors calculated the sample size. Please provide information.

5. In the discussion, the Authors write “However, the age of maternity was negatively associated with depression. It is therefore; likely, that mature woman can cope better with their decision to postpone pregnancy ”(p. 9, lines 331-333) - it is difficult to deny the obtained results, however, the results of research on the relationship between age and depression in mothers are inconclusive. Some of them point to the later age of the mother as a predictor of depression after childbirth. Therefore, it should be assumed that the results regarding the relationship between depression and age may depend on the presence or absence of other factors. Perhaps the constellation of sociodemographic variables in this particular study group is important. I think it is worth referring to in the discussion of the results.
6. The authors conclude: "In addition, pregnancy concerns and life satisfaction could be predictors of depression and resilience / optimism, respectively." (p. 11, lines 417-418) - indeed, the results show a relationship. However, I wonder if in the case of the "resilience" variable we should actually talk about the dependence in the direction indicated by the authors? Resilience is defined as a dynamic process reflecting the relatively good adaptation of an individual despite the threats or traumatic experiences experienced by him (cf. Luthar, 2006; Luthar & Zelazo, 2003). It seems that resilience will be a predictor of life satisfaction and pregnancy concerns, i.e. the higher the level of resilience, the higher the satisfaction with life and lower the concerns related to pregnancy. It seems to me that it is worth commenting on the obtained results, taking into account the bi-directional nature of the described relationship.

Luthar, SS. Resilience in development: A synthesis of research across five decades. In: Cicchetti, D.;Cohen, DJ., editors. Developmenthal Psychopatology: Risk, disorder, and adaptation. Vol. 2. Vol. 3.New York: Wiley; 2006. p. 740-795.8.

or

Luthar, SS.; Zelazo, LB. Research on Resilience. An Integrative Review. In: Luthar, SS., editor.Resilience and Vulnerability. Cambridge University Press; 2003. p. 510-549

Minor shortcomings:
1. Abbreviations should be explained the first time they are used in the body of the article. Please provide an extension of the abbreviation ART. (p.3, line 121)
2. The authors write about the use of LOT to measure optimism, but it is not explained what LOT is and how it relates to the variable optimism (p. 4, lines 160-161)
3. The authors write "We used the individual Spanish version of resilience" (p. 4, lines 167-168). I understand that they mean the Spanish Resilience Scale? Please clarify.

Author Response

Response: Thank you for taking your time to review our manuscript. Please, see below the responses and modifications parts in the article.

  1. The authors state that they applied “(...) two ad-hoc scales, family-work conflict, and pregnancy concerns” (p. 4, lines 172-174). While they quite accurately describe the structure and what is measured by the second tool created for the purposes of this study, the scale for measuring family-work conflict is described with little precision. Please provide more detailed description.

Response: More details about the purposes of the scale were added (page 4, section 2.3.3).

  1. Why were the variables entered into the regression equation by time of measurement? It seems more reasonalble to group the variables according to their nature, e.g. medical, related to reproductive experiences (ART, parity miscarriages ...).

Response: In this work, we aimed to test the evolution of the variables along pregnancy, which requires time-oriented regression models. This approach has been previously used in other clinical settings for follow up studies (PMID: 30160065), characterizing sets of antecedents/covariates by the time they occur or are identified. For this reason, we had introduced the variables sequentially.

  1. The authors state that: “The rate of twin pregnancies was 51.8% (...)" (p.5, lines 231-232) - one of the main variables representing the neonatal outcome is the child's birth weight. At the same time, more than half of the pregnancies are twin pregnancies in which the birth weight is usually lower. In this situation, it seems appropriate to perform a regression analysis separately for single and twin pregnancies.

Response: We agree on the need to consider the lower weight in twins. We did not analyze separately twin and single pregnancies, but we have taken into account this fact.  Due to the high collinearity of the use of ART with twin pregnancies (p-value=0.001), we decided to include ART in the models to evaluate the role of sociodemographic variables on the outcomes. We have modified the text to make it more comprehensive (lines 252-254).

  1. How did the Authors calculated the sample size. Please provide information.

Response: This was a non-interventional observational study. Since all women were healthy at baseline and we did not randomize to treatment arm, it was not necessary to determine the sample size a prior but we have established rigorous criteria for type I error (p-value).

  1. In the discussion, the Authors write “However, the age of maternity was negatively associated with depression. It is therefore; likely, that mature woman can cope better with their decision to postpone pregnancy” (p. 9, lines 331-333) - it is difficult to deny the obtained results, however, the results of research on the relationship between age and depression in mothers are inconclusive. Some of them point to the later age of the mother as a predictor of depression after childbirth. Therefore, it should be assumed that the results regarding the relationship between depression and age may depend on the presence or absence of other factors. Perhaps the constellation of sociodemographic variables in this particular study group is important. I think it is worth referring to in the discussion of the results.

Response: Very suitable this comment. We really appreciate it. We have added more information in the discussion section (lines 338-341 and reference 45). 

  1. The authors conclude: "In addition, pregnancy concerns and life satisfaction could be predictors of depression and resilience / optimism, respectively." (p. 11, lines 417-418) - indeed, the results show a relationship. However, I wonder if in the case of the "resilience" variable we should actually talk about the dependence in the direction indicated by the authors? Resilience is defined as a dynamic process reflecting the relatively good adaptation of an individual despite the threats or traumatic experiences experienced by him (cf. Luthar, 2006; Luthar & Zelazo, 2003). It seems that resilience will be a predictor of life satisfaction and pregnancy concerns, i.e. the higher the level of resilience, the higher the satisfaction with life and lower the concerns related to pregnancy. It seems to me that it is worth commenting on the obtained results, taking into account the bi-directional nature of the described relationship.
  • Luthar, SS. Resilience in development: A synthesis of research across five decades. In: Cicchetti, D.;Cohen, DJ., editors. Developmenthal Psychopatology: Risk, disorder, and adaptation. Vol. 2. Vol. 3.New York: Wiley; 2006. p. 740-795.8.
  • Luthar, SS.; Zelazo, LB. Research on Resilience. An Integrative Review. In: Luthar, SS., editor.Resilience and Vulnerability. Cambridge University Press; 2003. p. 510-549

Response: One again, very interesting comments, which we have considered. We have added more details in the discussion section (lines 385-390), modified the conclusions, and added reference 53.

Minor shortcomings:

  1. Abbreviations should be explained the first time they are used in the body of the article. Please provide an extension of the abbreviation ART. (p.3, line 121)

Response: the abbreviation was added for the first time in the text.

  1. The authors write about the use of LOT to measure optimism, but it is not explained what LOT is and how it relates to the variable optimism (p. 4, lines 160-161)

Response: the sentence was modified to facilitate comprehension.

  1. The authors write "We used the individual Spanish version of resilience" (p. 4, lines 167-168). I understand that they mean the Spanish Resilience Scale? Please clarify.

Response: the validated and adapted version of the resilience scale for the Spanish population was used. It is the same resilience scale published in 1993, but the version for the comprehension of the items for the Spanish-speaking population.

Reviewer 2 Report

Authors found Early pregnancy concerns and life satisfaction influence depression and resilience/optimism later on. Maternal resilience and life satisfaction was associated with birth weight. 

These findings are important and may be useful for the better maternal management.

Author Response

Response: Thank you for taking your time to review our manuscript. Studying pregnancy from a bio-psycho-social perspective, integrating all aspects of an individual's physical and mental health is important to understand factors associated with the disease.

Reviewer 3 Report

Maternal Resources, Pregnancy Concern, and Biological Factors Associated to Birth Weight and Psychological Health is an interesting assessment of maternal factors throughout pregnancy that affect neonatal birthweight.  There are some concerns, though.  

Introduction

This section could be improved by reworking the 3rd paragraph regarding the COR model.  As presented, the description is vague a difficult to follow.  Also, the last sentence could e confusing to readers, Consider different punctuation or splitting the sentence into two.  

Materials and Methods

  1. Were blood samples fasting?
  2. Section 2.3.1 needs to be revised.  It is a bit disorganized and some information is redundant. 
  3. Section 2.4 - it is a bit confusing when you say that psychological variables were collected at three timepoints because they all were not.  Respecifying here which were collected at each time would be helpful. 
    1. You indicate that outcomes are depression, resilience and optimism, but there is no indication of how depression was calculated.  Is it the depression portion of the HAD or the PANAS? 

Results

  1.  It would be beneficial to see the Ns for your groups.  For example, you indicate that 51.8 % of the pregnancies were twin? Given that rate, specifying birthweight by singleton vs multiple pregnancies is important, particularly since birthweight is one of your dependent variables. It may be that your effect of ART is really the result of multiple pregnancies.
  2. Table 4 - Birthweight

Discussion

  1. Alternately use Family-work and work-family.  
  2. Page 10, 2nd full paragraph, the authors say that pregnancy concerns at 24 weeks gestation are related to resilience at the end of pregnancy. I just don't see this relationship.
  3. I agree with the limitations presented.  Measures of dietary patterns, specific leukocyte populations would be beneficial. 

Author Response

Response: Thank you for taking your time to review our manuscript. Please, see below the responses and modifications parts in the article.

Introduction. This section could be improved by reworking the 3rd paragraph regarding the COR model.  As presented, the description is vague a difficult to follow.  Also, the last sentence could be confusing to readers, Consider different punctuation or splitting the sentence into two. 

Response: We agree with this comment and we have modified the suggested parts of the introduction for better comprehension (lines 63-73).

Materials and Methods. Were blood samples fasting?

Response: Yes, we have added this information in the text.

Section 2.3.1 needs to be revised.  It is a bit disorganized and some information is redundant.

Section 2.4 - it is a bit confusing when you say that psychological variables were collected at three timepoints because they all were not. Specifying here which were collected at each time would be helpful.

Response: We have revised section 2.3.1, added missing data, and excluding redundant information. The scales PANAS and HAD were re-written and we have added the psychological variables with the specific time points of measurement in section 2.4.

You indicate that outcomes are depression, resilience, and optimism, but there is no indication of how depression was calculated.  Is it the depression portion of the HAD or the PANAS?

Response: Yes, HAD (Hospital Anxiety and Depression test) consists of two-subscales (anxiety and depression). The anxiety test was answered in the first trimester and the depression scale at the end of pregnancy. We have described this in more detail in section 2.3.1.

Results. It would be beneficial to see the Ns for your groups.  For example, you indicate that 51.8% of the pregnancies were twin? Given that rate, specifying birthweight by singleton vs multiple pregnancies is important, particularly since birthweight is one of your dependent variables. It may be that your effect of ART is really the result of multiple pregnancies (Table 4 – Birthweight).

Response: We have included the sample size per relative frequency variables. We agree on the importance to consider differences in birth weight between single and twin pregnancies. We have taken this fact into account in our model. Due to the high collinearity of ART with twin pregnancies (p-value=0.001), we included ART in the models to assess the role of sociodemographic variables on the outcomes. We have modified the text to make it more comprehensive (lines 252-254).

Discussion. Alternately use Family-work and work-family. 

Response: Thank you. We have corrected this and used family-work throughout the text.

Page 10, 2nd full paragraph, the authors say that pregnancy concerns at 24 weeks gestation are related to resilience at the end of pregnancy. I just don't see this relationship.

Response: We agree with this assessment. Although Table 1 shows a negative correlation between pregnancy concerns and resilience, this correlation was not significant. We have indicated this connotation in the text (line 382).

I agree with the limitations presented.  Measures of dietary patterns, specific leukocyte populations would be beneficial.

Response: Thank you. Other spheres of pregnancy could explain part of the variance of the data.